# Histological E-data Registration in rodent Brain Spaces

**Jingyi Guo Fuglstad\*†§, Pearl Saldanha†, Jacopo Paglia#, Jonathan R Whitlock\*§**

Kavli Institute for Systems Neuroscience, Norwegian University of Science and Technology (NTNU), Trondheim, Norway

**Abstract** Recording technologies for rodents have seen huge advances in the last decade, allowing users to sample thousands of neurons simultaneously from multiple brain regions. This has prompted the need for digital tool kits to aid in curating anatomical data, however, existing tools either provide limited functionalities or require users to be proficient in coding to use them. To address this we created HERBS (Histological E-data Registration in rodent Brain Spaces), a comprehensive new tool for rodent users that offers a broad range of functionalities through a user-friendly graphical user interface. Prior to experiments, HERBS can be used to plan coordinates for implanting electrodes, targeting viral injections or tracers. After experiments, users can register recording electrode locations (e.g. Neuropixels and tetrodes), viral expression, or other anatomical features, and visualize the results in 2D or 3D. Additionally, HERBS can delineate labeling from multiple injections across tissue sections and obtain individual cell counts. Regional delineations in HERBS are based either on annotated 3D volumes from the Waxholm Space Atlas of the Sprague Dawley Rat Brain or the Allen Mouse Brain Atlas, though HERBS can work with compatible volume atlases from any species users wish to install. HERBS allows users to scroll through the digital brain atlases and provides custom-angle slice cuts through the volumes, and supports free-transformation of tissue sections to atlas slices. Furthermore, HERBS allows users to reconstruct a 3D brain mesh with tissue from individual animals. HERBS is a multi-platform open-source Python package that is available on PyPI and GitHub, and is compatible with Windows, macOS, and Linux operating systems.

**\*For correspondence:**
jingyi.guo@ntnu.no (JGF);
jonathan.whitlock@ntnu.no
(JRW)

†These authors contributed equally to this work

**Present address:** #KnowBe4, Apotekergata, Norway

§These authors jointly supervised this work

**Competing interest:** The authors declare that no competing interests exist.

## Editor's evaluation

This paper provides the field with a new and important Python-based tool to assist neurosurgery both before and after a wide range of interventions. In its present form, the software comes as a convincing toolbox that may be helpful for researchers relying on neurosurgery in rodents (both mice and rats).

## Introduction

Understanding the structure, function, or physiology of neural circuits requires the ability to consistently target brain regions prior to experiments and, afterward, to maintain an accessible, objective record of the areas studied (*Simmons and Swanson, 2009*). Traditionally, post hoc anatomical record-keeping has relied on histological approaches where users delineate anatomical features in tissue sections, then register them manually with a reference atlas. While these and other approaches are critical for grounding experiments anatomically, doing them has become increasingly challenging with the emergence of large-scale recordings spanning multiple brain regions.

Several software tools have been developed in recent years which use computer vision technology to accelerate and systematize the mapping of experimental results in various model species (e.g., *Shamash et al., 2018*; *Song et al., 2020*; *Claudi et al., 2021*; *Andy, 2022*). When combined with the

appropriate reference atlases (e.g., *Franklin and Paxinos, 2008*; *Wang et al., 2020*; *Claudi et al., 2020*), these tool kits give users the ability to reconstruct electrode tracks, viral expression, anatomical projections (*Oh et al., 2014*; *Zingg et al., 2014*; *Winnubst et al., 2019*), cell types (*Fürth et al., 2018*), or to detect cells (*Tyson et al., 2021*), gene expression patterns, or functional nodes (*Lein et al., 2007*; *Ortiz et al., 2020*) in the brain.

Different types of tool kits have been developed for different brain preparations, including volumetric whole-brain data or sliced tissue sections, with each approach bringing its own strengths and limitations. Whole-brain volume analyses have the advantage of readily conveying the 3D spatial relationships between neuronal pathways and neighboring structures. For instance, ITK-SNAP (*Yushkevich et al., 2006*) is a popular biomedical software used for automatic image segmentation and delineation of regions of interest in human brain imaging data. For animal models, there is *Brainrender* (*Claudi et al., 2021*), which displays any of several atlases incorporated from the BrainGlobe Atlas API in 3D. It provides fast visualization for user-defined data sets, such as reconstructed electrode tracks, viral expression, or anatomical projections. These can be obtained when *Brainrender* is combined with sibling software, Brainreg (*Tyson et al., 2020*), which maps whole-brain data sets to atlases using supervised automatic image registration. While whole-brain data sets provide a holistic quantitative dimension to analyses, generating them is time and resource intensive, requiring either a light sheet scanning microscope or magnetic resonance imaging.

Because of the comparatively lower cost and simplicity, anatomical mapping using histological tissue sectioning still dominates in most labs (e.g., *Clancy et al., 2019*; *Zutshi et al., 2022*; *Lagartos-Donate et al., 2022*; *Gardner et al., 2022*). This, however, also has its own challenges. One common issue lies in matching a given tissue slice correctly to a reference atlas template. Previous studies have sought to solve this problem using multiple software platforms, like ImageJ (*Schneider et al., 2012*), Adobe Illustrator (*Adobe Inc, 2023b*), or Adobe Photoshop (*Adobe Inc, 2023a*), which can be time consuming for manual curation and is not systematic. Software packages have been developed to address this issue, including the electrode reconstruction tool kit, SHARP-TRACK (*Shamash et al., 2018*). This software uses a global warping transformation method to fit histological images back to a mouse atlas. Afterward, it provides a read-out of the estimated number of channels in each brain region as well as 3D visualizations. As currently configured, the tool kit works only with mouse atlases and is MATLAB-based, which requires a paid license and some programming knowledge.

Despite the number of tool kits available, there currently is no single software that combines functionalities spanning the gamut of pre-surgical planning and post-experiment registration without requiring programming. To help fill this methodological gap, we developed HERBS, an open-source Python package which brings together a range of functionalities in a user-friendly graphical front-end interface. HERBS can generate pre-surgical coordinates, reconstruct electrode tracks after recordings, mark viral expression volumes, or single cells; delineate regional boundaries in brain slices, and view post hoc anatomical reconstructions in 2D or 3D. There is no upper limit for the number of electrodes that it can reconstruct, so it can be used with microelectrode arrays or multi-tetrode bundles. To serve both rat and mouse users, HERBS has the in-built function to download and run the Waxholm Space (WHS) Atlas of the Sprague Dawley Rat Brain (*Papp et al., 2014*) or the Allen Mouse Brain Atlas (*Wang et al., 2020*). The software also contains an atlas explorer that can scroll freely through either rat or mouse atlases to display coronal, sagittal, and horizontal planar surfaces in 2D, as well as the corresponding mesh or volume of the atlas in 3D.

For accurate and robust image registration, HERBS provides rigid registration as well as interactive, non-rigid registration using triangulation-based piece-wise affine transformation. It is compatible with most tissue staining methods and image formats, as long as the regions of interest are visible and can be delineated by the user. For users who wish to augment or add functionalities of their own in Python, we provide a standalone Python pipeline tool kit, TRACER (Toolkit for Reconstructing Anatomical CoorindatEs in Rats), in parallel with HERBS (see Materials and methods for more information). To maximize general accessibility, HERBS requires minimal programming knowledge and is compatible with Windows, macOS, and Linux systems. It is intended as an open community resource available for further expansion and refinement. We hope it offers a valuable addition to the anatomical methods available for the neuroscience community.

## Results

### General outline

HERBS was designed as an intuitive, generic software tool to provide rodent users with four core functionalities, shown in *Figure 1A*. These include (a) generating anatomical target coordinates when planning a surgery; (b) processing and editing raw histological data; (c) registering and reconstructing objects implanted in the brain; and (d) visualizing annotated brain volumes in 3D. Users can carry out each functionality by following a series of steps described in the HERBS Cookbook using a keyboard and mouse. The layout of the user interface for HERBS, including color-coded panels used to execute its functionalities, are shown in *Figure 1B*.

When installing HERBS, users can choose whether to use the WHS Atlas of the Sprague Dawley Rat Brain (*Papp et al., 2014*) or the Allen Mouse Brain Atlas (*Wang et al., 2020*) as the anatomical reference (see Cookbook, chapter 6.1). Either atlas can be downloaded through HERBS, as they are both integrated as parts of the software package. We point users wishing to employ other volumetric atlases, including for other species, to section 6.1.3 of the HERBS CookBook for links to illustrated instructions on uploading and installing additional atlases.

### Generating pre-surgical coordinates

One of the essential steps in planning a surgery is defining the stereotaxic coordinates to reach specific brain areas of interest. HERBS facilitates this process by allowing users to determine the insertion parameters required to target one or several regions along a linear trajectory. This is done by simply clicking the start point and end point of the desired path on a 2D brain slice image, and there is no limit to the number of trajectories (e.g. recording probes) that can be planned. HERBS then automatically computes the insertion angle and length needed to reach the target brain region, as well as the location of the trajectory relative to Bregma defined in the atlas. The user can print and bring the read-out to the surgery room or save for record keeping. The planning functionality is equally applicable for targeting imaging lenses, injection pipettes for tracer or virus injections, or making focal lesions.

The working steps for generating pre-surgical coordinates are illustrated in *Figure 2*, in which we plan the insertion of a multi-shank Neuropixels 2.0 probe through retrosplenial cortex, visual cortex, and the intermediate hippocampus in the right hemisphere of a rat. The step-by-step protocol is described in the HERBS Cookbook, Chapter 6.4. Neuropixels users should note that estimates of probe length include the 175-µm channel-less shank tip, and that HERBS eliminates it when estimating the number of recording channels in the brain. In the first step, the user scrolls to the brain slice of interest (*Figure 2A*). This is aided by the color coding of each brain area in the atlas, which provides visual landmarks for defining boundaries between regions. Next, the probe trajectory is defined in the region(s) of interest with clicks of the mouse (*Figure 2B*). Once completed, the probe trajectory is saved as a layer in the Side Bar, after which it is saved to a 3D representation showing the trajectory of the probe in a WHS brain volume (*Figure 2C*). After the probe is merged on the Side Bar, a read-out is generated for a 4-shank probe (*Figure 2D*) that includes the location and regions traversed. A more detailed read-out table for each shank of the 4-shank probe can also be made (*Figure 2E*) which includes inclination, length of probe inserted, number of recording channels per region, and entry and exit coordinates of the probe. Either or both read-outs can be saved per the user's desire. As there is no limit to the number of probes that can be planned for insertion, the user simply repeats the steps above for each probe. Users can also apply this functionality to target multiple regions using multi-probe implants with user-defined spacing and geometries. Users can define the recording site layout on individual probes as well (see section 6.6 of the CookBook for instructions and links to illustrated tutorials).

### Reconstructing electrode tracks

Another core functionality of HERBS is reconstructing electrode paths after surgeries or in vivo recordings, and there is no upper limit on the number of electrodes that can be reconstructed. Prior to reconstructing probe trajectories, histological images can be pre-processed before they are overlaid on their corresponding atlas images. Pre-processing can include rotating the section to an optimal plane, changing brightness, gamma curves, and so forth, and can be done using the image view

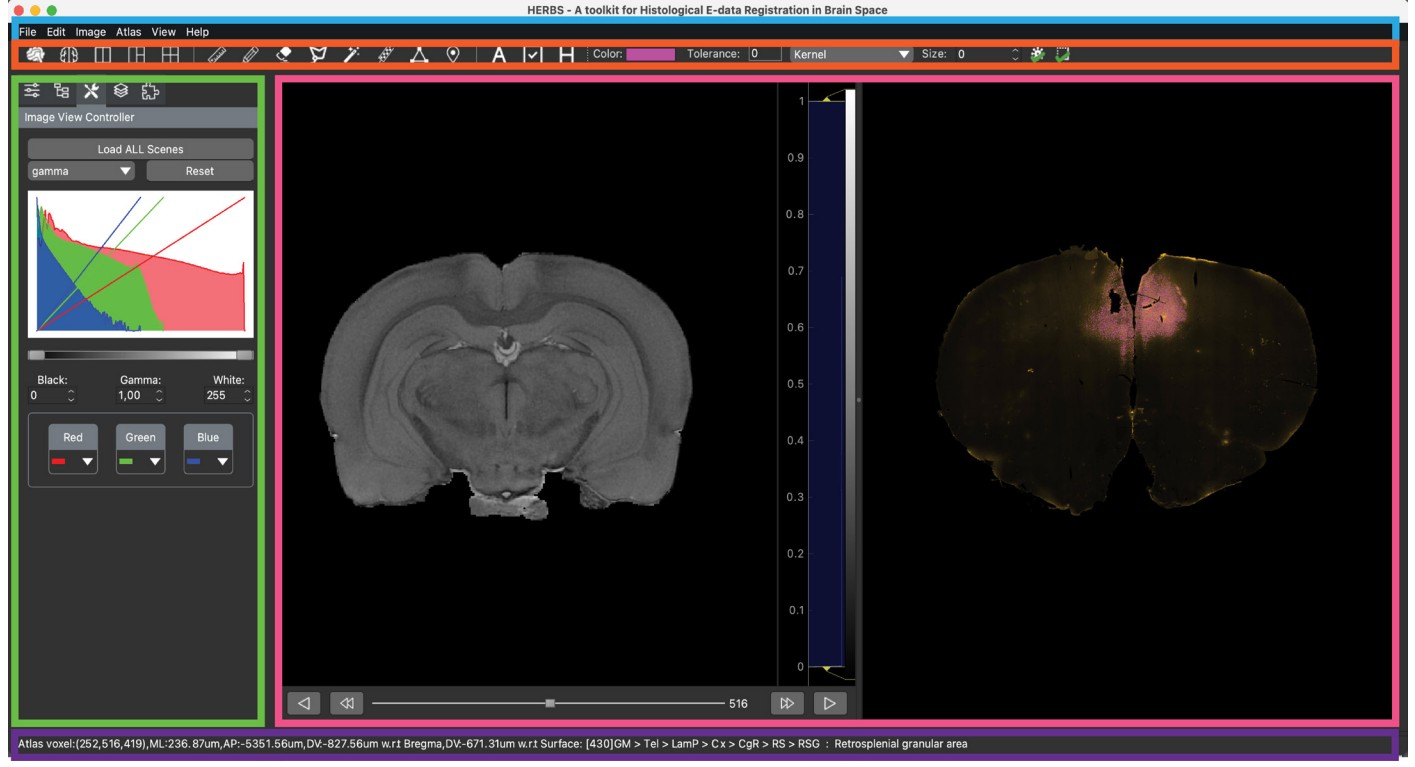

**Figure 1.** The core functionalities and user interface of HERBS. (**A**) HERBS can be used to (**a**) plan electrode insertions or viral injections before surgery, (**b**) process and edit histological images, (**c**) register objects such as probes, virus, cells, and slice boundaries from histological images, and (**d**) visualize the brain in 3D with defined functions. (**B**) The user interface of HERBS consists of the Menu Bar (outlined in blue) for navigating through the functions, the Tool Bar (outlined in red) from which users select actions to perform in the Plot Window (magenta), such as registering or reconstruct objects. The Side Bar (geen) provides tools for visually editing objects or layers. The Status Bar (in purple at bottom) shows the brain regions and coordinates displayed in the Plot Window. The WHS or Allen Mouse Brain atlases can be downloaded and loaded directly through HERBS from the Menu Bar with a click of a button.

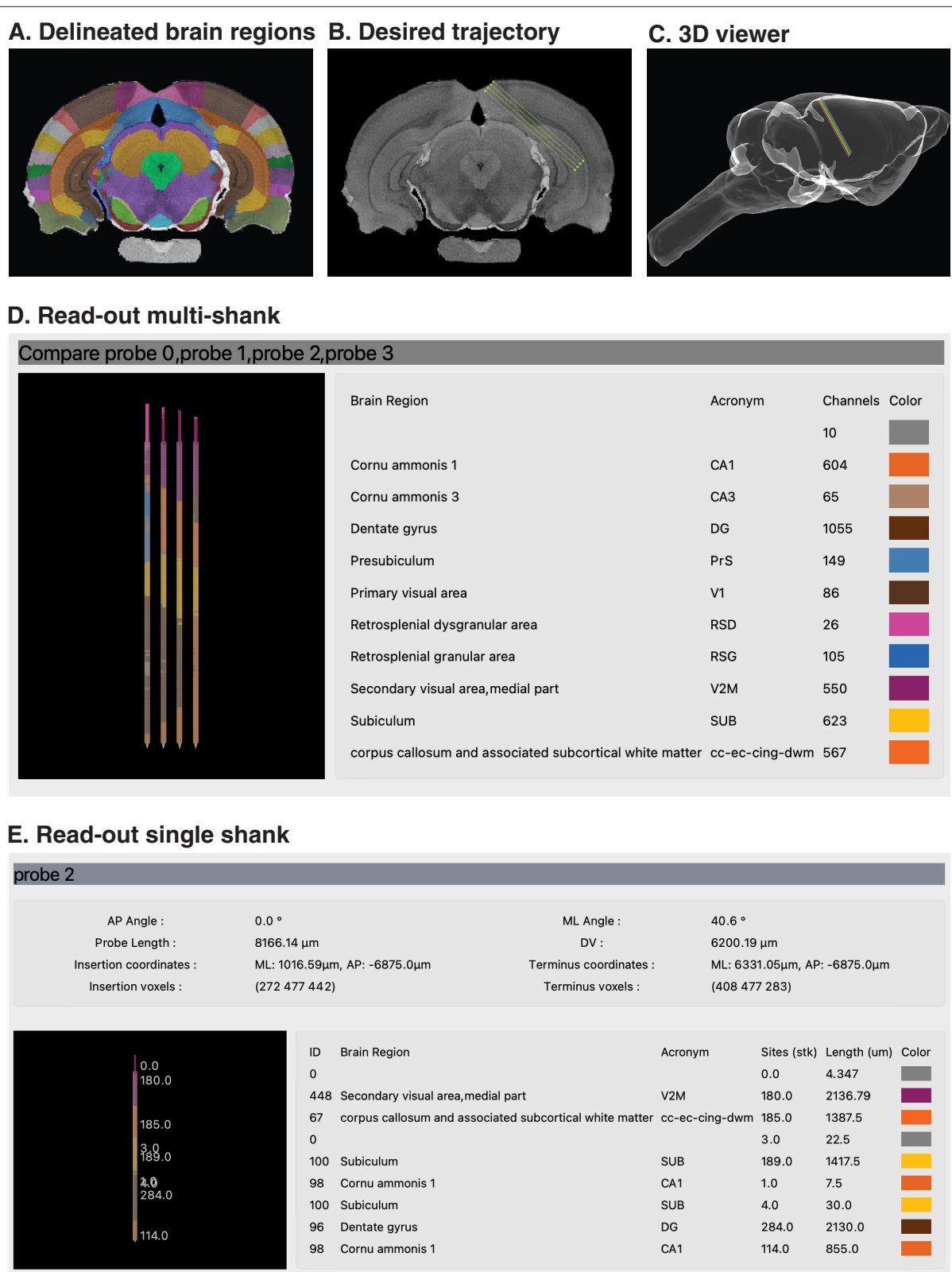

**Figure 2.** Using HERBS to plan a Neuropixels 2.0 probe implant before surgery. (**A**) Once the plane of choice is set (coronal in this example), 2D atlas brain sections with regional delineations appear and the user scrolls to the slice with the region(s) they wish to target. (**B**) The user defines the trajectory of the probe by clicking the desired start and end-points (the insertion angle here is 40.8° lateral tilt at Bremga –6.8 mm). (**C**) A linear probe trajectory is projected into the Waxholm Space in 3D. (**D**) A read-out table for all four probes shows the regions the probe has traversed and the number of channels

*Figure 2 continued*

in each region. (**E**) A detailed read-out for a single probe provides the entry position, probe angle, and insertion length to reach the target regions, as well as an estimation of the number of recording channels in each region in the experiments.

controller in the Side Bar, or the Edit option in the Menu Bar, as indicated in the interface panel in *Figure 1B* (see HERBS Cookbook, chapter 6.6).

After the image has been loaded and adjusted, the user can begin the steps of the reconstruction pipeline shown in *Figure 3*, which uses an example section with a track from a DiI-stained Neuropixels 1.0 probe (red) progressing laterally from visual to auditory cortex (an example image for probe reconstruction is provided in *Figure 3—figure supplement 1*). The first step is choosing the plane of sectioning (coronal, sagittal, and horizontal) and scrolling to the corresponding atlas brain slice. Once the corresponding slice is located, the user defines anchor points to warp the histological section onto the atlas section (*Figure 3A*) (see Cookbook, chapters 6.4 and 6.5). Once the anchor points are defined and the matching step is complete, the user then clicks a minimum of 4 points along the probe track in the tissue, after which a 3D rendering is generated showing how the probe is situated in the volume of the brain (*Figure 3B*) (Cookbook, chapter 6.6.1). A detailed read-out table showing the inclination and length of the probe, regions traversed and number of recording channels per region are generated automatically (*Figure 3C*). In cases where histological sections are cut out of plane, HERBS allows the user to tilt the atlas brain up to 30° to facilitate template matching. We also note that electrode track reconstructions can be performed for any type of electrodes, tetrodes, or recording arrays, as well as with different sizes of sections. HERBS facilitates this by allowing users to build user-defined geometry and channel layouts of linear silicon probes (see the subfolder "HERBS/image/probe_related" on the HERBS GitHub page, linked in "Data availability, software, and citation policy" below).

## Reconstruction of viral expression

Beyond electrode reconstructions, HERBS can be used for anatomical applications including visualization of virus expression, tracer labeling, or marking lesions or neurodegeneration. Here, we provide a brief overview of the procedure for reconstructing a spatial volume of adenovirus (AAV) expression (see Cookbook, chapter 6.7 for this and related protocols). The procedure for reconstructing viral expresion volumes is similar to that for probe reconstruction, but the regions defined in each slice are two-dimensional, and the volume to delineate will likely extend over multiple slices (*Figure 4*). As with reconstructing a probe track, the first step in reconstructing a viral volume is matching the starting histology section with the corresponding slice in the atlas brain and overlaying the two. The initial slice from an example viral reconstruction is shown in *Figure 4A*, and the same image is provided for user practice in *Figure 4—figure supplement 1*. The user then defines the perimeter of virus expression by selecting the magic wand tool from the Tool Bar and clicking on a fluorescent region of tissue in the histological section (*Figure 4B*). Note that the granularity of the selected region will depend on the magic wand tolerance set by the user. These steps are repeated for each slice included in the visualization. Finally, the marked fluorescent regions from several 2D sections are combined and projected into a 3D brain volume (*Figure 4C*), and the list of brain regions in contact with the virus are shown in the read-out with color-coded labels (*Figure 4D*). The same functions and steps are used for similar tasks such as labeling single cells, marking fibers of passage, or visualizing lesions. Note also that the resolution of the 3D reconstruction will depend on how many sections the user includes in the reconstruction; the example in *Figure 4C* consisted of 11 tissue slices. It is also possible to load cell point data generated from other programs, such as CellFinder (*Tyson et al., 2021*), and to visualize those data in 3D brain volumes. Instructions for importing external cell point data can be found in section 6.8.3 of the HERBS Cookbook.

## Discussion

With HERBS, we seek to furnish rodent users with a user-friendly, click-button GUI that simplifies the process of curating histological data. The functionalities include planning coordinates ahead of surgeries as well as registering and visualizing anatomical data in 2D or 3D after experiments are completed. The GUI can be used to delineate a range of features such as recording locations, lesions

**A. Reconstructing the electrode track**

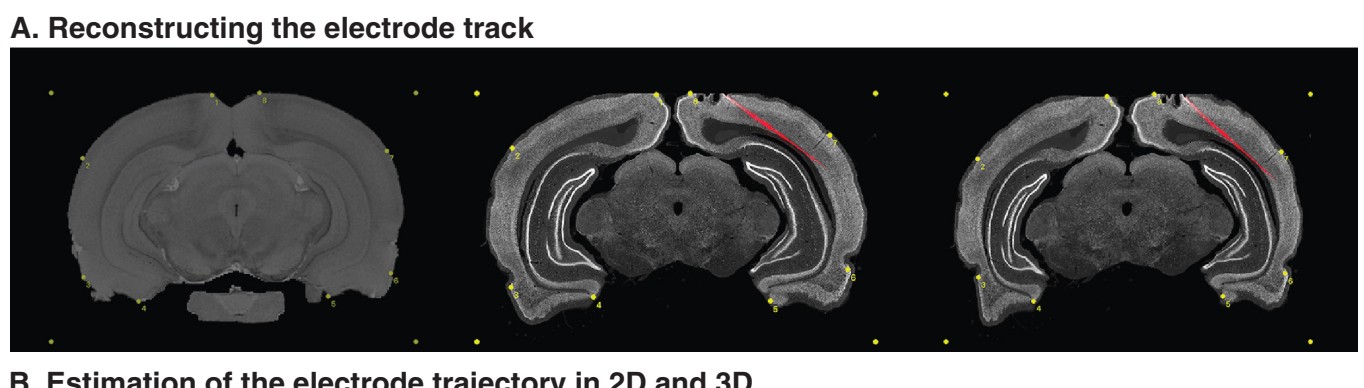

**B. Estimation of the electrode trajectory in 2D and 3D**

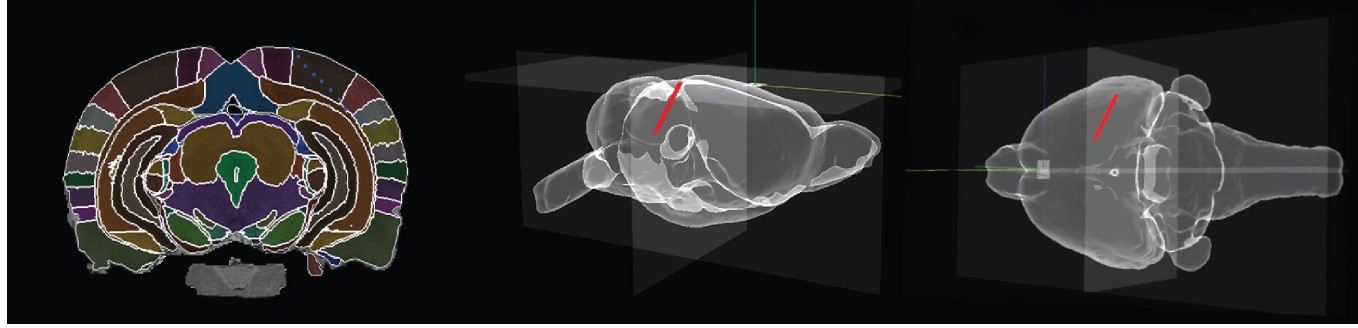

**C. Read-out of the reconstruction process**

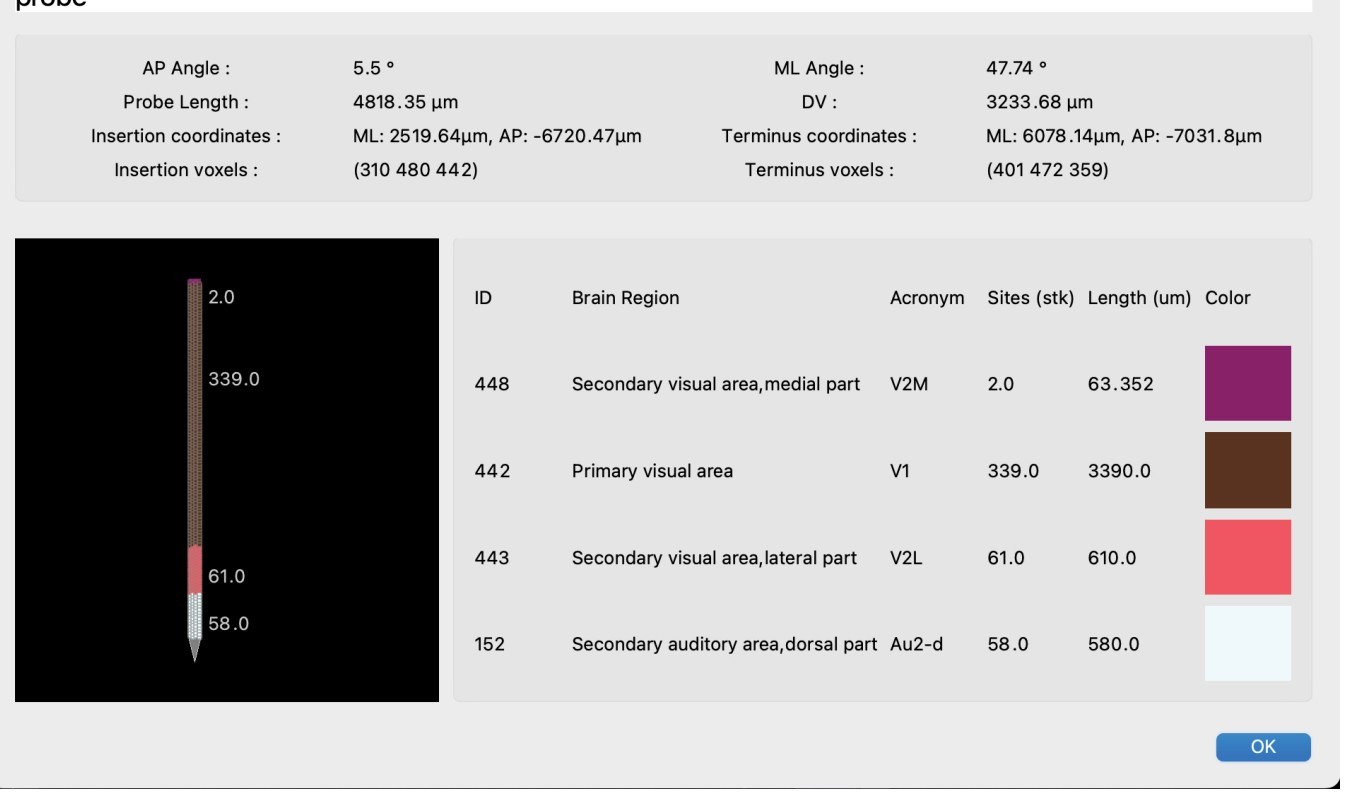

probe

| | | | |
|---|---|---|---|
| AP Angle : | 5.5 ° | ML Angle : | 47.74 ° |
| Probe Length : | 4818.35 μm | DV : | 3233.68 μm |
| Insertion coordinates : | ML: 2519.64μm, AP: -6720.47μm | Terminus coordinates : | ML: 6078.14μm, AP: -7031.8μm |
| Insertion voxels : | (310 480 442) | Terminus voxels : | (401 472 359) |

| ID | Brain Region | Acronym | Sites (stk) | Length (um) | Color |
|---|---|---|---|---|---|
| 448 | Secondary visual area, medial part | V2M | 2.0 | 63.352 | |
| 442 | Primary visual area | V1 | 339.0 | 3390.0 | |
| 443 | Secondary visual area, lateral part | V2L | 61.0 | 610.0 | |
| 152 | Secondary auditory area, dorsal part | Au2-d | 58.0 | 580.0 | |

OK

**Figure 3.** Reconstructing a recording probe trajectory from histological sections. (**A**) The initial steps include scrolling to the atlas brain slice (left) that best matches the user's histological section (middle). User-defined anchor points are then used to stretch and conform the histological section onto the atlas section; once they are merged a new image is generated with the histology warped to overlay the atlas slice (right panel). (**B**) The user then generates an estimate of probe placement and length by clicking at least four points spanning from the entry of the probe in the brain to the distal

*Figure 3 continued on next page*

*Figure 3 continued*

tip (left). The 3D reconstructed placement of the probe is shown in the sagittal plane (middle panel) and viewed from the top (right panel) in the WHS volume. (**C**) A read-out from the merged probe shows details of the probe insertion, the coordinates, as well as the brain regions the probe traversed in different colors. WHS, Waxholm Space.

The online version of this article includes the following figure supplement(s) for figure 3:

**Figure supplement 1.** An example image for learning to use the probe reconstruction functionality, as shown in *Figure 3*.

or viral expression and is compatible with potentially any histological staining method. With HERBS, the user marks regions of interest on tissue images directly and each slice is mapped to a template, which should greatly reduce subjective judgement and variability across samples and brains. For rat users, HERBS references the newest version (v4) of the WHS rat brain atlas (*Papp et al., 2014*), which contains over 200 labeled cortical and sub-cortical regions at sub-millimeter scales, and for mouse users it references the widely-adopted Allen Mouse Brain Atlas (*Wang et al., 2020*).

## Novelty and utility of HERBS in relation to other tool kits

Although several software packages with similar functionalities have been released in recent years (see *Supplementary file 1*), there are a number of features that distinguish HERBS from them. Foremost is its simplicity, since it works via a visually intuitive, click-button interface designed to minimize

**A. Virus expression in tissue**  **B. Tissue warped to atlas**  **C. 3D visualization of virus**

**D. Read-out for areas containing viral expression**

## virus

| ID | Brain Region | Acronym | Volume (stk voxel) | Proportion (%) | Color |
|---|---|---|---|---|---|
| 0 | | | 1 | 0.0 | |
| 67 | corpus callosum and associated subcortical white matter | cc-ec-cing-dwm | 5 | 0.0 | |
| 405 | Prelimbic area | PrL | 4930 | 1.79 | |
| 406 | Secondary motor area | M2 | 3692 | 0.4 | |
| 408 | Primary motor area | M1 | 33 | 0.0 | |
| 411 | Cingulate area 1 | Cg1 | 6187 | 2.41 | |

**Figure 4.** 3D visualization of virus expression across multiple tissue sections. (**A**) Shows the spatial extent of AAV-mediated mCherry expression in a coronal section from the frontal cortex of an adult rat. (**B**) The region of viral expression is defined by clicking the fluorescent region of tissue with the magic wand tool; areal boundaries are included at this stage. (**C**) A volumetric rendering of the brain shows the extent of viral expression in cingulate cortex and surrounding areas, with the volume of virus spread (based on the 11 slices used here) expressed in voxels, and the proportion of each region expressing the virus show as a percentage. (**D**) A read-out of the regions in which the virus was expressed.

The online version of this article includes the following figure supplement(s) for figure 4:

**Figure supplement 1.** An example image for learning to reconstruct viral expression in a single brain slice, as shown in *Figure 4*.

user effort, and it requires minimal coding and no added software. It is also unique in that it brings together both pre- and post-surgical registration and mapping functionalities in one program. The fact that it specifically includes rats also stands out among the many high-quality rendering and digital atlas resources for mice, such as Brain Mesh (*Yaoyao-Hao, 2020*), SHARPTRACK (*Shamash et al., 2018*), cocoframer (*Lein et al., 2007*), and others (e.g. *Jin et al., 2019*, *Chon et al., 2019*).

Another recently available anatomical tool kit is *Brainrender* (*Claudi et al., 2021*), which produces visually appealing anatomical renderings from a variety of source atlases through the BrainAtlasglobe API, but differs from HERBS in important ways. The most essential difference is that *Brainrender* was designed as a *visualization* tool. As such, it renders anatomical data in 3D volumes, but it cannot register anatomical data in a reference atlas on its own. Registration must be done beforehand with separate software such as Brainreg (*Tyson et al., 2022*) or CellFinder (*Tyson et al., 2021*). HERBS, in contrast, was designed primarily as a *registration* tool, which comes configured to work with primary source atlases for rats or mice. The HERBS GUI can both download and load these reference atlases, whereas *Brainrender* relies on separate software and plugins which require programming knowledge in Python. The output from HERBS could be used as input for *Brainrender* if formatted appropriately, though HERBS offers some of its own visualization functionalities for simplicity. Finally, Brainreg and *Brainrender* use whole-brain data from light sheet imaging, whereas HERBS works with commonly used histological tissue sectioning methods.

The Allen Institute recently released the Neuropixels Trajectory Explorer (*Andy, 2022*.), a MATLAB-based platform which can be used to plan coordinates for electrophysiological probes prior to surgery in mice and rats. Currently, the pre-surgical registration step is the only functionality available, as it does not yet have the option to load histology for post hoc processing of probe tracks or viral expression. Unlike *Brainrender* and HERBS, it runs in MATLAB, which typically requires a paid software license. Thus, the main features that distinguish HERBS from other tools are that it (i) offers a combination of functionalities found separately in *Brainrender* and Neuropixels trajectory explorer, (ii) all functionalities in HERBS are accessible with button clicks and keystrokes. It includes reconstructing probe tracks of neuropixel 1.0 and 2.0, tetrodes and linear silicon probes (with user defined geometry and channel layout); along with registering virus expressed with a read-out of the spread in brain regions in volume and percentage, and (iii) it can be used immediately with the WHS Rat Brain Atlas or the Allen Mouse Brain Atlas without requiring additional plugins. Beyond these features, HERBS, similar to Brainreg (*Tyson et al., 2022*), can incorporate other atlases given the correctly formatted files, such as the Paxinos and Watson atlases for rats or mice, provided the user has the appropriate licensing permissions.

## Sources of error and future improvements

While we hope that HERBS proves useful in its current form, there are areas where it can be improved and expanded in the future. For example, the template matching step can be time consuming if data sets contain a large number of histological slices, as with virus expression or tracer labeling. We aim to streamline these processes by implementing machine vision-based algorithms for pre-processing user images and matching them to template slices automatically in future versions of HERBS. Furthermore, although we designed HERBS to minimize the need for programming, installing it could be challenging for users who have not installed Python before.

In developing HERBS we also encountered sources of error which appear general to working with anatomical tissue that has been preserved with fixatives. For instance, we found that probe length estimates can be less consistent when larger numbers of histological slices are used for reconstructions. We suspect these errors could arise from averaging across clicked points defining the probe trajectory, and it is possible that misalignment of clicked points between slices can add to this. Another issue is that fixing tissue with paraformaldehyde causes non-uniform shrinking of the tissue. We found this could lead to small discrepancies when comparing Neuropixels channel counts from HERBS, which uses fixed tissue, against LFP measures taken while the animals were still alive. Although tissue shrinkage is inherent to histological processing and cannot be removed mathematically, HERBS offers a solution by fitting warped tissue to the atlas template using multiple anchor points per slice, which should substantially reduce inconsistencies due to shrinkage.

Accurately registering raw neuroanatomical data in a reference atlas framework, though critically important, is time consuming, tedious and variable from user to user. HERBS was developed to

expedite, simplify, and systematize this process by providing a single tool for generating anatomical coordinates, as well as annotated 2D and 3D visualizations and data tables summarizing its outputs. We wish to emphasize that HERBS is a completely open-source software with which we aim to help our fellow research community. We therefore welcome users to submit suggestions for improvements and to report bugs on our GitHub page, as this will help us continually improve the software and user experience.

# Materials and methods

**Key resources table**

| Reagent type (species) or resource | Designation | Source or reference | Identifiers | Additional information |
|---|---|---|---|---|
| Software, algorithm | Numpy | *Harris et al., 2020* | RRID:SCR_008633 | |
| Software, algorithm | Scipy | *Virtanen et al., 2020* | RRID:SCR_008058 | |
| Software, algorithm | OpenCV | *Bradski, 2000* | RRID:SCR_01552 | |
| Software, algorithm | PyOpenGL | *Woo et al., 1999* | | |
| Software, algorithm | PyQT5 | *PyQT, 2012* | | |

## Python packages and resources

HERBS is written entirely in Python version three using basic Python packages, such as numpy, opencv, and scipy, as mentioned in the table below. Details on the specific dependencies used in HERBS can be found in HERBS Cookbook (chapter 3). Documentation and step-wise instructions for installing and implementing HERBS can be found in the HERBS Cookbook (chapter 4). HERBS is an open-source software with hard-coded Python scripts. Should a user wish to customize Python-pipeline, one can use the code of TRACER—which is open-source and available at the Github repository (*Paglia et al., 2021*). To use HERBS, we recommend users to create a PyCharm project with Python installation (version 3.8.10) and other supporting packages which are specified in the dependencies section in the HERBS Cookbook (chapter 4).

## Neuropixels and virus expression

All experiments were performed in accordance with the Norwegian Animal Welfare Act and the European Convention for the Protection of Vertebrate Animals used for Experimental and Other Scientific Purposes. All procedures were approved by the Norwegian Food Safety Authority (Mattilsynet; protocol IDs 27175 and 25094). All tissue for in-house testing came from adult (>15 weeks) Long-Evans hooded rats. Detailed steps of the surgical preparation and post-operative care are described in *Mimica et al., 2018*.

### Neuropixels

The probes (version 1.0 Neuropixels, IMEC, Belgium) were coated with DiI (Vybrant DiI, catalog no. V22888, Thermo Fisher Scientific, USA) by repeatedly drawing a 2 µL droplet of DiI solution at the tip of a micropipette up and down the length of the probe shank until all DiI had dried onto the shank, causing the shank to appear pink. The probes were angled in the arm of a stereotaxic frame and inserted at a rate of 100–300 µm per min. The chronically implanted animal was kept <1 week following surgery, after which it was perfused with 4% paraformaldehyde and the brains were removed.

The brain was transferred to 2% dimethyl sulfoxide (DMSO; VWR, USA) solution for cryoprotection for 1–2 days, after which the Neuropixel shank was removed, the brain was frozen and sectioned in 50 µm slices with a sliding microtome (Microm HM430, Thermo Fisher Scientific). The tissue sections then underwent fluorescent immunostaining against NeuN (catalog no. ABN90P, Sigma-Alrich, USA), followed by secondary antibody-staining with Alexa 647-tagged goat anti-guinea pig antibody (catalog no. A21450, Thermo Fisher Scientific), after which the sections were rinsed, mounted, cover-slipped, and stored at $4^0$C. The detailed immunostaining protocol is available per request. Next, the sections were digitized using an automated scanner for fluorescence and brightfield images at the

appropriate illumination wavelengths (Zeiss Axio Scan.Z1, Jena, Germany), and were saved in .jpeg format for processing in TRACER.

## Virus expression

Pulled glass pipettes were used to inject 500 nL of AAV5-mDlx-Chr2-mCherry-Fishell-3 (plasmid no. 83898, Addgene, USA; AAV produced at Kavli Viral Vector Core Facility, NTNU) at a rate of 50 nL per min. Bilateral injections were targeted to the cingulate cortex (Cg1; +2.0 AP, ±0.5 ML, –2.0DV). Five weeks after surgery, the animal was killed and perfused with 4% paraformaldehyde. The brain was removed, post-fixed overnight in 4% paraformaldehyde at $4^0$C, then cryoprotected for 24 hr in 2% DMSO. The brain was then frozen in dry ice and 40 µm sections were collected as described above. Fluorescent signal from the virus was amplified by immunostaining against Red Fluorescent Protein (catalog no. 5F8, Chromotek GmbH, Germany), followed by secondary antibody-staining with Alexa 546-tagged Goat anti-rat IgG (catalog no. A-11081, Thermo Fisher Scientific). The detailed immunostaining protocol is available upon request.

## Data availability, software, and citation policy

The software described in this manuscript is an open-source software written completely in Python. HERBS is fully supported by Windows, macOS, and Linux. Source code is available on https://github.com/Whitlock-Group/HERBS (*Fuglstad, 2022*). HERBS Cookbook and documents are available on the Whitlock group Github page.

The WHS rat brain atlas files can be found here on the NITRC website (https://www.nitrc.org/projects/whs-sd-atlas). The required atlas files to run HERBS (downloaded upon installation of HERBS) and TRACER are cited below:

WHS rat brain atlas v4:

WHS_SD_rat_atlas_v4.nii.gz (Kleven et al., in preparation)
WHS_SD_rat_atlas_v4.label (Kleven et al., in preparation)

Previous version of the WHS rat brain atlas (v1.01):

WHS_SD_rat_T2star_v1.01.nii.gz (*Papp et al., 2014*)
WHS_SD_rat_brainmask_v1.01.nii.gz (*Papp et al., 2014*)

## Citation policy

We kindly ask users to cite this paper when using HERBS or TRACER in their studies, and to cite the appropriate version of the WHS rat brain atlas.

Refer to the WHS Atlas by its RRID: SCR_017124, and cite the first publication (*Papp et al., 2014*) along with the version of the atlas that is used. For example, cite *Kjonigsen et al., 2015*, v2 of the atlas if the user's work makes particular use of delineations of the hippocampal region. Or cite *Osen et al., 2019*, v3 of the atlas if the user's work makes particular use of delineations of the auditory system.

## License

The WHS Atlas of the Sprague Dawley rat brain is licensed under the Creative Commons Attribution ShareAlike (CC BY-SA) 4.0 license: https://creativecommons.org/licenses/by-sa/4.0/.

The Allen Mouse Brain Atlas software and wiki are freely available at https://github.com/cortex-lab/allenCCF (*Shamash et al., 2022*).

## Acknowledgements

The authors thank T B Leergaard, I E Bjerke, and H Kleven for graciously sharing and assisting with the Waxholm Space atlas version 4; T Tombaz, B Mimica, E H Holmberg, and I Rautio for sample tissue; M Andresen and GM Olsen for technical assistance; E H Holmberg, B Kanter, T Tomaz, I Rautio and A Vollan, J Rudolf, D Hayden for beta testing HERBS and/or TRACER functionalities; M P Witter, GM Olsen, and members of the Whitlock lab for helpful discussion; B Mimica and T Tombaz for motivating the need for such a tool kit. This work was supported by a Research Council of Norway

FRIPRO Grant (no. 300709) to JRW, the Centre of Excellence Scheme of the Research Council of Norway (Centre for Neural Computation, Grant no. 223262), the National Infrastructure Scheme of the Research Council of Norway – NORBRAIN (Grant no. 197467), and The Kavli Foundation. The MRI core facility is funded by the Faculty of Medicine at NTNU and Central Norway Regional Health Authority.

## Additional information

### Funding

| Funder | Grant reference number | Author |
| --- | --- | --- |
| Norges Forskningsråd | 300709 | Jonathan R Whitlock |
| Norges Forskningsråd | 223262 | Jonathan R Whitlock |
| Norges Forskningsråd | 197467 | Jonathan R Whitlock |
| Kavli Foundation | | Jonathan R Whitlock |

The funders had no role in study design, data collection and interpretation, or the decision to submit the work for publication.

### Author contributions

Jingyi Guo Fuglstad, Conceptualization, Data curation, Software, Supervision, Validation, Visualization, Writing – review and editing; Pearl Saldanha, Software, Validation, Visualization, Writing - original draft, Writing – review and editing; Jacopo Paglia, Software, Validation, Visualization; Jonathan R Whitlock, Conceptualization, Supervision, Funding acquisition, Project administration, Writing – review and editing

### Author ORCIDs

Pearl Saldanha http://orcid.org/0000-0002-6749-8240
Jonathan R Whitlock http://orcid.org/0000-0003-2642-8737

### Ethics

All experiments were performed in accordance with the Norwegian Animal Welfare Act and the European Convention for the Protection of Vertebrate Animals used for Experimental and Other Scientific Purposes. All procedures were approved by the Norwegian Food Safety Authority (Mattilsynet; protocol IDs 27175 and 25094). All tissue for in-house testing came from adult (>15wk) Long-Evans hooded rats. Detailed steps of the surgical preparation and post-operative care are described in Mimica et al. 2018 (doi:10.1126/science.aau2013).

### Decision letter and Author response

Decision letter https://doi.org/10.7554/eLife.83496.sa1
Author response https://doi.org/10.7554/eLife.83496.sa2

## Additional files

### Supplementary files

• MDAR checklist

• Supplementary file 1. Table summarizing the currently available anatomical tool kits for use with rodents. The name and year of publication for each tool kit are shown at top, and the functionalities of each tool kit are listed below.

### Data availability

The software described in this manuscript is an open-source software written completely in Python 3.8. HERBS is fully supported by Windows, macOS and Linux. Source code, HERBS Cookbook and documentation are available on the Whitlock group Github page: https://github.com/Whitlock-Group/HERBS. The Waxholm Space rat brain atlas files can be found here from the NITRC website:

https://www.nitrc.org/projects/whs-sd-atlas. The Allen Mouse Brain Atlas software and wiki are freely available at: https://github.com/cortex-lab/allenCCF.

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
