## [Editor Report]

This paper provides the field with a new and important Python-based tool to assist neurosurgery both before and after a wide range of interventions. In its present form, the software comes as a convincing toolbox that may be helpful for researchers relying on neurosurgery in rodents (both mice and rats).

---

## [Decision Letter]

**Decision letter after peer review:**

[Editors’ note: the authors submitted for reconsideration following the decision after peer review. What follows is the decision letter after the first round of review.]

Thank you for submitting the paper "TRACER: A toolkit to register and visualize anatomical coordinates in the rat brain" for consideration by *eLife*. Your article has been reviewed by 3 peer reviewers, including Mathieu Wolff as the Reviewing Editor and Reviewer #3, and the evaluation has been overseen by a Senior Editor.

Comments to the Authors:

We are sorry to say that, after consultation with the reviewers, we have decided that this work will not be considered further for publication by *eLife*.

In short, all reviewers found the tool described of this study with a clear value but it really needs further polishing to be widely used within the neuroscience community. At present the installation process is by itself an obstacle for any user with only moderate familiarity with Python-based environment. So this would need to be considerably enhanced to consider a new examination of this tool. In addition a more thorough description of its specific value versus other tools already available including for rats is warranted to understand the added value that TRACER brings.

*Reviewer #1 (Recommendations for the authors):*

General comments:

It seems like a lot of work to go through all sections and match to the digital atlas. Incorporating matching between pre-processed images and reference sections seems crucial for widespread use of the tool. If this is something that cannot be done in short order, a widespread use of the tool may be limited.

More extensive comments should be made about Brainrender, which was published in *eLife* in March 2021. This tool is only mentioned and no extensive comparison with TRACER is made.

Software issues:

In evaluating the package (on a machine with Linux Mint 20 Cinnamon, version 4.6.7. Linux Kernel, 5.4.0-89-generic), there were issues with getting the atlas to load. It was an issue with an xcf library, so perhaps this is very specific to this setup, but perhaps is worth noting that some Linux users may have a similar issue.

More generally, there were issues with getting the latest atlas (v4) to work. An error was received that the process was "Killed" and then a message stating that the atlas v3 couldn't be found. Downloading atlas v3 resolved the issue.

It would be useful to know if this software allow for localization of microelectrode arrays? The paper states that TRACER can place trajectory of 6 probes for prediction or reconstruction, so does this mean it can only be used for single point trajectories (such as those from single shank electrodes, fiber optic cannula, or a drug infusion cannula)? A statement should be included about the utility of TRACER for array probes. (If it is not capable of mapping arrays, perhaps this is a modification to consider for the future).

The text in Figure 5 as a bit small and maybe the axis labels could also be shortened and bounding boxes removed.

Potential issue with tissue processing:

How was shrinkage dealt with in processing the brain tissue? Was any estimate done? This might matter for the specific experiment reported in the manuscript, which involved staining with several antibodies. This could also be a major issue for users, especially in cases where histochemical or immunohistochemical stains might be combined in a study.

*Reviewer #2 (Recommendations for the authors):*

Strengths

– The tool and user manual both appear excellent. The authors have put a lot of effort into making this tool.

– Prior tools were focused on mice, while this one is optimized for rat histology.

– The "surgery planning" aspects were interesting and very useful, particularly for deciding how many recording sites would end up in various brain regions with different insertion angles, etc.

Weaknesses

– There were files from the Waxholm Space rat brain atlas that needed to be downloaded. I clicked on the provided link but it wasn't clear how to find the relevant files, they appear to come from multiple different releases. Unless it violates licensing, can these be automatically included in TRACER? Extra steps like this can confuse and trip up new users (like me!), and potentially create obsolete instructions if the external links change their organization or content.

– The authors correctly note that installing and running TRACER requires some knowledge of Python and the command line. I see this as the biggest weakness of the package. They say they are working on a stand-alone GUI to run it, but in the meantime I suggest they develop instructional videos to talk a new user through a standard installation and example case.

– Both myself and a grad student in our lab tried to install the tool but ultimately failed to do so. We are moderately experienced with Python and we worked through some errors and felt like we were getting closer, but after ~3 hours we finally gave up. While I'm sure the fault is on our end to some degree, we were motivated to test the tool out and we are experienced in Python but we could not get it installed on our system. I suggest the author's β test the tool with novice users with no instruction other than what is on the Github, and see if they can discover any pain points in the installation process. I believe this is addressable, but is a critical weakness of the tool as it currently stands.

Suggestions for improving the user experience

– Strongly encourage the authors to upload the package to Pypi or conda package repository. This will make the installation of the dependencies and the package itself much easier.

– At the bare minimum, instructions should be expanded and clarified starting from the creation of the environment. If the terminal will be used anyway for installing the dependencies, it is much easier to create the environment using the Anaconda prompt from scratch.

– Downloading packages from Github can be confusing for people who are not experienced with this. Where the instructions say they need "TRACER package downloaded on your local computer…" I think the user manual should explicitly walk them through each step – What does a user do when they're looking at the Github page? Click download as Zip? Does it matter where this is saved? Should this be unzipped? Getting tripped up on these early steps can be deal breakers for new users.

– In the user manual, some screenshots would be helpful to orient the user, instead of things like "open the terminal by clicking the arrow near the name of your environment".

– After downloading the zip file from Github and extracting the package, the name of the folder is TRACER_main (or TRACER_master) and not just TRACER. So when trying to run it on Spyder (or wherever), it doesn't find the module.

– To run the package, the working directory has to be inside the first TRACER_main folder, since there is another TRACER_main folder inside the first. This tripped us up for a while.

Reviewer #3 (Recommendations for the authors):

The authors propose here a new open-source, python-based toolkit to reconstruct the trajectories of recording electrodes in the rat brain. Other possible applications include to visualize virus spread or to provide candidates stereotaxic coordinates before starting a surgery. At first glance, the tool is promising and may effectively fill a gap as many existing tools have been designed for mice primarily although rats continue to be highly relevant for behavioral studies. Installing and using the tool is not trivial for users with no Python experience though and the added value by comparison with other tools recently developed is unclear as detailed below.

Installing/running TRACER

The authors acknowledge in the short discussion that using TRACER may be challenging for users with no experience in Python-based environments. I can concur with that comment. I have asked a few trainees in the lab (with no coding experience) to try to install it and none of them was able to go through the entire process alone. So I think getting a GUI as stand-alone is really needed to impact on the field. Otherwise it is hard to appreciate why this could be more useful than any other tools. I would really encourage the authors to develop this GUI early on rather than only suggesting it's a possible future direction.

Added value versus other existing tools

Earlier this year, another paper was published at *eLife* documenting Brainrender, which is potentially suitable across species, including rats (Claudi et al., 2021). There are multiple other resources that are also available. I think a greater effort to explain how the present tool is positioned by respect with these other options is needed. This is very important for the field; while we can appreciate the value of having diverse tools to rely on, there is also some merit in adopting standards and splitting the community around multiple tools may not be beneficial in the long run. I think a much more thorough discussion is needed to present the PROs and CONs of TRACER versus Brainrender and other tools.

To address my general comments, I think the authors should really answer the following question: Considering that Brainrender is available in multiple species, what is the added value of TRACER?

I do not intend at all to minimize the work produced by the authors, I just want to make my point clearer from the end-user viewpoint: as many potentially equivalent tools exist, why choosing a specific one over the others?

[Editors’ note: further revisions were suggested prior to acceptance, as described below.]

Thank you for resubmitting your work entitled "HERBS: Histological E-data Registration in rodent Brain Spaces" for further consideration by *eLife*. Your revised article has been evaluated by Laura Colgin (Senior Editor) and a Reviewing Editor.

The manuscript has been improved but there are some remaining issues that need to be addressed, as outlined below:

All reviewers recognized the value of HERB to process brain histology data and think it could be particularly useful for people working with rats, for which very little tools are currently available. The reviewers also identified a number of points that, if adequately addressed, may considerably improve the impact of HERB for the field. The authors are encouraged to implement these points or at least to provide a thorough discussion on their will and ability to carry further developments of the software in the future.

1) There is a need for clearer plans for long term sustainability of the software following future releases of Python. The authors should indicate with more details if they have the manpower necessary to ensure the maintenance of all packages in the long run, and possibly to implement further features (see other points below)

2) Being able to incorporate other brain atlases and other packages (e.g. CellFinder) would enhance its value and the potential for HERB to be widely adopted.

3) Viral reconstruction could benefit from a more quantitative assessment: since the user is provided with a list of brain structures affected by the viral spread, it would be incredibly helpful to provide the volume estimates (or %) affected for each of these brain regions.

4) Automatic registration of sections is not available, and therefore requires extensive manual work to use the software. Perhaps the authors have plans to improve on this front.

*Reviewer #2 (Recommendations for the authors):*

We gave HERBS some testing in the team and we found that it could really be very helpful for many researchers.

One potential caveat will be to ensure that maintenance of the main packages is being covered as new versions of Python are being developed. Recommending older Python versions for HERB may not be a realistic option in the long run as this is currently suggested in the Cooking book (which currently do not point on the right page for Python 3.9.9 – p.10, section 3.4.2).

We do have a major suggestion for viral spread reconstruction as we think this particular option may be particularly useful. Since the user is provided with a list of brain structures affected by the viral spread, it would be incredibly helpful to provide the volume estimates (or %) affected for each of these brain regions. This would thus open the way for relatively unbiased quantitative analyses (specificity of viral spread, but also lesions, or any infusion). This is much needed in the field to encourage researchers to better report histology data and I strongly believe it will really make HERB reaching another level.

*Reviewer #3 (Recommendations for the authors):*

It would be useful to specify whether this toolkit covers the following features, and if so, how?

– Designing multi-area implants: does the first section (generating surgical coordinates) include the possibility to design multi-probe implants that could target different brain areas? this can be a great value to those who'd like prototype headstages for targeting multiple areas.

– Adaptable to other atlases: is it possible to upload brain atlas for similar animals (e.g. naked mole rats or tree shrews)? How about other available rat brain atlases?

– Integrating other available packages: it would be great if one could easily integrate e.g. CellFinder with HERBS.

---

## [Author Response]

[Editors’ note: the authors resubmitted a revised version of the paper for consideration. What follows is the authors’ response to the first round of review.]

Reviewer #1 (Recommendations for the authors):General comments:It seems like a lot of work to go through all sections and match to the digital atlas. Incorporating matching between pre-processed images and reference sections seems crucial for widespread use of the tool. If this is something that cannot be done in short order, a widespread use of the tool may be limited.

We agree that automatic template matching would be an extremely convenient functionality to include. We explored this possibility but found that it will require substantial development of machine vision algorithms to automatically match input images to a template. It is a difficult computer vision problem for several reasons, including (i) that histological input images can be stained in various ways which differ qualitatively from atlas images; (ii) the input and atlas images can be of different resolutions; (iii) input images can be slightly out of plane relative to the atlas. After extensive use of the tool, we find that users are likely the best judges of a match between input tissue and the atlas template. Automatic registration is a longer-term goal but one which we are not able to include in this version of the software.

More extensive comments should be made about Brainrender, which was published in eLife in March 2021. This tool is only mentioned and no extensive comparison with TRACER is made.

The new manuscript now describes *Brainrender* more fully in the Introduction (3^rd^ paragraph) and in the Discussion (3^rd^ paragraph), where we describe the main features which distinguish HERBS from *Brainrender*. This is part of a larger section in the Discussion, “Novelty and utility of HERBS in relation to other tool kits”, in which we discuss HERBS in relation to the other anatomical software released in recent years. As noted above, the primary difference between HERBS and *Brainrender* is that HERBS provides the core function of registering histological images back to a reference atlas, which gives reconstructions anatomical coordinates, in millimetres, relative to Bregma and the brain surface. *Brainrender* provides beautiful visualizations, but it cannot be used to register anatomical data, which is done in a separate step with separate software and plugins. This brings up another major difference, HERBS was designed to be self-sufficient—the GUI can be used to download and load its own source atlases, as well as register and visualize anatomical data without additional software programs.

Software issues:In evaluating the package (on a machine with Linux Mint 20 Cinnamon, version 4.6.7. Linux Kernel, 5.4.0-89-generic), there were issues with getting the atlas to load. It was an issue with an xcf library, so perhaps this is very specific to this setup, but perhaps is worth noting that some Linux users may have a similar issue.

We thank the Reviewer for pointing out this difficulty in Linux. We have refined the packaging of the Atlas and have been able to load it on Linux systems (Kubuntu and Ubuntu 18.04) subsequently. We hope this resolves the issue.

More generally, there were issues with getting the latest atlas (v4) to work. An error was received that the process was "Killed" and then a message stating that the atlas v3 couldn't be found. Downloading atlas v3 resolved the issue.

We have revamped the atlas downloading procedure by having a copy of the Waxholm Space Atlas (v4) ready for download from the GitHub repository. Both it and the Allen Mouse Brain Atlas can be downloaded in a click-button process when HERBS is installed. Details on how to do this are provided in the user manual (“HERBS Cookbook” Chapter 6.1) and the “ReadMe” tab on the GitHub page (https://github.com/Whitlock-Group/HERBS/tree/main/Tutorial).

It would be useful to know if this software allow for localization of microelectrode arrays? The paper states that TRACER can place trajectory of 6 probes for prediction or reconstruction, so does this mean it can only be used for single point trajectories (such as those from single shank electrodes, fiber optic cannula, or a drug infusion cannula)? A statement should be included about the utility of TRACER for array probes. (If it is not capable of mapping arrays, perhaps this is a modification to consider for the future).

We are glad to report that HERBS, unlike TRACER, has no upper limit on the number of probes / trajectories it can reconstruct, so it is compatible with multi-electrode arrays and multi-tetrode drives. Each trajectory (electrode) still needs to be defined individually by clicking the start- and endpoints, but again there is no upper bound on how many times this can be done. We note this feature of HERBS in the Introduction (top paragraph on Page 2), and in the Results sections, “Generating pre-surgical coordinates” (page 4, paragraph 2), and “Reconstructing electrode tracks” (page 6, paragraph 1).

The text in Figure 5 as a bit small and maybe the axis labels could also be shortened and bounding boxes removed.

Figure 5 from the original draft has been removed from the current version of the paper.

Potential issue with tissue processing:How was shrinkage dealt with in processing the brain tissue? Was any estimate done? This might matter for the specific experiment reported in the manuscript, which involved staining with several antibodies. This could also be a major issue for users, especially in cases where histochemical or immunohistochemical stains might be combined in a study.

This is an excellent question since tissue shrinkage is an inherent constraint when using any kind of tissue fixation, and it may be further compounded during immuno/histochemical staining protocols. While there is no way to regularize or remove shrinkage artifacts completely, our solution to deformation artifacts in HERBS is to include a “matching” step in which histological images are fit onto the reference atlas template. Specifically, HERBS uses triangulation-based piecewise affine transformation to locally warp the histological image to fit the template, based on key point the user defines with mouse-clicks. The warping and template matching methods are noted in the Introduction (2^nd^ paragraph on page 2) and in Results section, “Reconstructing electrode tracks” (2^nd^ paragraph on page 6). The stepwise process for matching is laid out in detail and with illustrations in the HERBS CookBook (Chapter 6.5).

Reviewer #2 (Recommendations for the authors):Strengths– The tool and user manual both appear excellent. The authors have put a lot of effort into making this tool.– Prior tools were focused on mice, while this one is optimized for rat histology.– The "surgery planning" aspects were interesting and very useful, particularly for deciding how many recording sites would end up in various brain regions with different insertion angles, etc.

We thank the Reviewer for their praise and are happy that the amount of effort that went into TRACER was apparent when it was reviewed. We are also glad to share that the new HERBS software now works for both rats and mice, and we note that the surgery planning functionalities are included in the present tool kit as well.

Weaknesses– There were files from the Waxholm Space rat brain atlas that needed to be downloaded. I clicked on the provided link but it wasn't clear how to find the relevant files, they appear to come from multiple different releases. Unless it violates licensing, can these be automatically included in TRACER? Extra steps like this can confuse and trip up new users (like me!), and potentially create obsolete instructions if the external links change their organization or content.

We are sorry for the previous difficulty in locating and downloading the relevant files for the Waxholm Space rat brain atlas. We have revamped the atlas downloading procedure by including file links in HERBS so that either the Waxholm Space Rat Brain Atlas (v4) or the Allen Mouse Brain Atlas (Wang et al., *Cell*, 2020) can be downloaded in a click-button process after HERBS is installed. Details on how to do this are provided in the user manual (“HERBS Cookbook”, chapter 6.1) and the “Tutorial” tab on the GitHub page (https://github.com/Whitlock-Group/HERBS/tree/main/Tutorial).

– The authors correctly note that installing and running TRACER requires some knowledge of Python and the command line. I see this as the biggest weakness of the package. They say they are working on a stand-alone GUI to run it, but in the meantime I suggest they develop instructional videos to talk a new user through a standard installation and example case.

We have radically simplified the process of installing HERBS, in that it requires a single command (“pip install herbs”) in a Python terminal. However, this assumes the user is already running a current version of Python, and installation of Python can be challenging for someone who has never done so. To make the process easier, we include links to download the appropriate versions of Python (3.8.10 or 3.9.9) in Chapter 3 of the HERBS Cookbook, and in Chapter 4 we give instructions for installing HERBS from PyPI or GitHub. As for running HERBS, all functionalities are performed with mouse clicks and keystrokes and do not require extensive programming knowledge.

– Both myself and a grad student in our lab tried to install the tool but ultimately failed to do so. We are moderately experienced with Python and we worked through some errors and felt like we were getting closer, but after ~3 hours we finally gave up. While I'm sure the fault is on our end to some degree, we were motivated to test the tool out and we are experienced in Python but we could not get it installed on our system. I suggest the author's β test the tool with novice users with no instruction other than what is on the Github, and see if they can discover any pain points in the installation process. I believe this is addressable, but is a critical weakness of the tool as it currently stands.

We again apologize for the difficulty in installing and running TRACER and are very sorry for costing the Reviewer’s and trainee’s time. To avoid this happening with HERBS, we have had multiple β testers vet the downloading and installation process on their own per the written instructions provided in the Cookbook.

Suggestions for improving the user experience– Strongly encourage the authors to upload the package to Pypi or conda package repository. This will make the installation of the dependencies and the package itself much easier.

This is an excellent suggestion, and we are happy to share that HERBS has been uploaded to PyPI: https://pypi.org/project/herbs/

– At the bare minimum, instructions should be expanded and clarified starting from the creation of the environment. If the terminal will be used anyway for installing the dependencies, it is much easier to create the environment using the Anaconda prompt from scratch.

We understand and apologize for the prior lack of clear instructions. We sought to address this issue with additional installation instructions in the HERBS Cookbook, Chapter 4. All useful links for the installation of anaconda and Conda environments have been included in this section. We believe we have solved the installation of dependencies in making HERBS a PyPI package.

– Downloading packages from Github can be confusing for people who are not experienced with this. Where the instructions say they need "TRACER package downloaded on your local computer…" I think the user manual should explicitly walk them through each step – What does a user do when they're looking at the Github page? Click download as Zip? Does it matter where this is saved? Should this be unzipped? Getting tripped up on these early steps can be deal breakers for new users.

We thank the reviewer for pointing out these pitfalls for new users. We have radically simplified the process of installing HERBS by uploading it to PyPI, and by including instructions on downloading HERBS from GitHub, both on the ReadMe page and in the HERBS cookbook. Specifically, Chapter 3 of the Cookbook includes links to download the appropriate versions of Python (3.8.10 or 3.9.9), and Chapter 4 includes basic installation instructions for HERBS (which are single commands) from PyPI and GitHub.

– In the user manual, some screenshots would be helpful to orient the user, instead of things like "open the terminal by clicking the arrow near the name of your environment".

We thank the Reviewer for this helpful suggestion and again apologize for the difficulties with the user manual for TRACER. Accordingly, the HERBS Cookbook contains numerous screenshots and directions for every step of the core functionalities of the software (the “HERBS recipes” in Chapter 6). We sincerely hope for a better user experience with the HERBS Cookbook.

– After downloading the zip file from Github and extracting the package, the name of the folder is TRACER_main (or TRACER_master) and not just TRACER. So when trying to run it on Spyder (or wherever), it doesn't find the module.

We are very sorry for the difficulties faced for the installation of the TRACER software.

We have sought to solve installation-related technical issues by making HERBS a PyPI package which can be installed in the terminal or a Python IDE with a simple line. Chapter 4 of the CookBook includes these instructions on the installation process.

– To run the package, the working directory has to be inside the first TRACER_main folder, since there is another TRACER_main folder inside the first. This tripped us up for a while.

The new installation process for HERBS solves this issue as well.

Reviewer #3 (Recommendations for the authors):The authors propose here a new open-source, python-based toolkit to reconstruct the trajectories of recording electrodes in the rat brain. Other possible applications include to visualize virus spread or to provide candidates stereotaxic coordinates before starting a surgery. At first glance, the tool is promising and may effectively fill a gap as many existing tools have been designed for mice primarily although rats continue to be highly relevant for behavioral studies. Installing and using the tool is not trivial for users with no Python experience though and the added value by comparison with other tools recently developed is unclear as detailed below.

We thank the Reviewer for taking the time to evaluate and give feedback on the previous submission and TRACER tool kit. Based on the difficulties reported in downloading and operating TRACER in Python, we have re-vamped the tool to work as a click-button GUI, and only one command line in Python is required for the initial installation. Beyond that, the functionalities of HERBS are driven with a mouse and keyboard following detailed instructions, with illustrations, in the Cookbook. We also discuss in more detail how HERBS compares against and differs from other software released in recent years, and have included a new Supplementary Table comparing functionalities of the different tool kits.

Installing/running TRACERThe authors acknowledge in the short discussion that using TRACER may be challenging for users with no experience in Python-based environments. I can concur with that comment. I have asked a few trainees in the lab (with no coding experience) to try to install it and none of them was able to go through the entire process alone. So I think getting a GUI as stand-alone is really needed to impact on the field. Otherwise it is hard to appreciate why this could be more useful than any other tools. I would really encourage the authors to develop this GUI early on rather than only suggesting it's a possible future direction.

We apologize for difficulties the Reviewer and their trainees encountered in downloading and using TRACER previously, and we thank them for their time and effort. These and similar sentiments from the other Reviewers prompted us to produce the new HERBS GUI in the current paper, which we hope is far simpler to sue. We could not remove usage of Python from the process completely, but once the tool is installed and running it requires no further programming or command windows. We hope the tool in its current form gives a better experience and will reach a wider user base.

Added value versus other existing toolsEarlier this year, another paper was published at eLife documenting Brainrender, which is potentially suitable across species, including rats (Claudi et al., 2021). There are multiple other resources that are also available. I think a greater effort to explain how the present tool is positioned by respect with these other options is needed. This is very important for the field; while we can appreciate the value of having diverse tools to rely on, there is also some merit in adopting standards and splitting the community around multiple tools may not be beneficial in the long run. I think a much more thorough discussion is needed to present the PROs and CONs of TRACER versus Brainrender and other tools.

We fully agree that it is important for the paper to clearly state what functionalities HERBS and other software offers, and what differentiates HERBS from the tools already out there. This can be found in a new section in the Discussion entitled “Novelty and utility of HERBS in relation to other tool kits”. As noted at the beginning of the Rebuttal, we feel the key distinguishing points are (i) the diversity of functionalities it offers (pre- and post-surgical), (ii) the simplistic, click-button interface of the GUI, and (iii) that the relevant mouse or rat reference atlases can be downloaded from a drop-down menu in the GUI—so it does not require other software. Moreover, it works with nearly any type of slice histology or staining method. These and other details are listed in the new Supplementary Table in the revised manuscript. We hope the HERBS GUI will be of service to those with little to no coding experience.

To address my general comments, I think the authors should really answer the following question: Considering that Brainrender is available in multiple species, what is the added value of TRACER?

We can absolutely appreciate this question, not only with respect to *Brainrender*, but other tools released in the last year. As noted above in the Rebuttal, *Brainrender* is an outstanding visualization resource, but it cannot on its own perform anatomical registration to a reference atlas. This must be done with separate software and plugins, which requires knowledge of programming that a proportion of would-be users may lack. HERBS, on the other hand, was conceived as a registration software that comes with its primary reference atlases in tow, so it fulfills a complementary role to *Brainrender’s* visualization functions. We also included some 2D and 3D visualization functions in HERBS to reduce the need for added software and steps, but we do not include animations like *Brainrender*.

I do not intend at all to minimize the work produced by the authors, I just want to make my point clearer from the end-user viewpoint: as many potentially equivalent tools exist, why choosing a specific one over the others?

We thank the Reviewer for considering our effort and agree it is important to clarify which tools do what, and how the present tool kit is different. We hope the revised manuscript does this more effectively than before. After the original TRACER manuscript was reviewed and rejected, we took the Reviewer's input seriously and sought to generate a radically simpler tool that minimizes user effort. Our main objective with HERBS was to make a tool that we would want to use in our own lab to expedite the time-consuming process of anatomical registration, especially across subjects.

[Editors’ note: what follows is the authors’ response to the second round of review.]The manuscript has been improved but there are some remaining issues that need to be addressed, as outlined below:All reviewers recognized the value of HERB to process brain histology data and think it could be particularly useful for people working with rats, for which very little tools are currently available. The reviewers also identified a number of points that, if adequately addressed, may considerably improve the impact of HERB for the field. The authors are encouraged to implement these points or at least to provide a thorough discussion on their will and ability to carry further developments of the software in the future.1) There is a need for clearer plans for long term sustainability of the software following future releases of Python. The authors should indicate with more details if they have the manpower necessary to ensure the maintenance of all packages in the long run, and possibly to implement further features (see other points below)

The creator of HERBS, Jingyi Guo Fuglstad, has and will continue to actively respond to technical queries posted on the HERBS GitHub page. The co-first author, Pearl Saldanha, will regularly check for updates on dependencies and new versions of Python. HERBS and the CookBook will be duly updated after major updates in Python are released.

So far, in response to GitHub requests and discussions, several new features have already been implemented, including compatibility with different kinds of linear silicon probes (the geometry and site layout of which the user can define), unmerging of object layers, as well as other functions described below.

2) Being able to incorporate other brain atlases and other packages (e.g. CellFinder) would enhance its value and the potential for HERB to be widely adopted.

We agree that compatibility with other brain atlases would enhance the utility of HERBS for a broader user base. As detailed below in response to Reviewer 3, HERBS is now compatible with any brain atlas, so long as the user can provide the appropriate volume files, segmentation files, and label information. HERBS can also use Mask files if they are included with a given atlas (e.g. the Waxholm Space Atlas). Illustrated instructions on how to upload and process other atlases can be found on the HERBS GitHub Tutorial sub-paged (https://github.com/Whitlock-Group/HERBS/blob/main/Tutorial/upload_and_process_user_defined_atlases.md). This link is noted and included in subsection 6.1.3 of the HERBS CookBook, and the added functionality is stated in the Abstract and on lines 79-82 of the General Outline section in the Results.

As further requested, we made it possible to load cell point data generated from CellFinder, and to visualize those data in 3D brain volumes according to whichever atlas the user has installed. Instructions on how to do this can be found on the HERBS GitHub sub-page (https://github.com/Whitlock-Group/HERBS/blob/main/Tutorial/External_Related/upload_external_cell_points.md), and can be found in the 6.8.3 sub-section of the CookBook. This is also noted in the manuscript on lines 161-165 of the “Reconstruction of viral expression” sub-section in the Results.

3) Viral reconstruction could benefit from a more quantitative assessment: since the user is provided with a list of brain structures affected by the viral spread, it would be incredibly helpful to provide the volume estimates (or %) affected for each of these brain regions.

We agree this would be highly beneficial for users wishing to have an unbiased estimate of viral spread in affected brain areas. Accordingly, we have updated the viral reconstruction functionality to include the volume (in voxels) of virus expressed in each brain region, as well as the estimated proportion of that brain region expressing the virus. We note that this applies the same for any kind of tissue labeling or lesions in a region of interest.

The newest version of HERBS generates these estimates automatically for viral expression analyses, and an example of the updated output is shown in the revised Figure 4D in the manuscript (new columns in the readout are for “Volume (stk voxel)” and “Proportion (%)” of each area containing viral expression or labeling).

4) Automatic registration of sections is not available, and therefore requires extensive manual work to use the software. Perhaps the authors have plans to improve on this front.

We acknowledge that it would great to add automated registration and that it would expedite the workflow in HERBS. However, adding automatic slice registration is a very large request, since doing so properly would require the development of sufficiently robust machine vision algorithms that can handle variable histological input. We respectfully suggest that developing such algorithms falls outside the scope of the current version of HERBS, which we feel nevertheless brings considerable added value to users wishing register histological slices in rodents or other species. Automated registration is part of our long-term strategy for HERBS, and this is noted in lines 218-220 of the revised manuscript.

Reviewer #2 (Recommendations for the authors):We gave HERBS some testing in the team and we found that it could really be very helpful for many researchers.One potential caveat will be to ensure that maintenance of the main packages is being covered as new versions of Python are being developed. Recommending older Python versions for HERB may not be a realistic option in the long run as this is currently suggested in the Cooking book (which currently do not point on the right page for Python 3.9.9 – p.10, section 3.4.2).

Please see our response to this same query in Point (1).

We do have a major suggestion for viral spread reconstruction as we think this particular option may be particularly useful. Since the user is provided with a list of brain structures affected by the viral spread, it would be incredibly helpful to provide the volume estimates (or %) affected for each of these brain regions. This would thus open the way for relatively unbiased quantitative analyses (specificity of viral spread, but also lesions, or any infusion). This is much needed in the field to encourage researchers to better report histology data and I strongly believe it will really make HERB reaching another level.

Please see our response to this same query in Point (3).

Reviewer #3 (Recommendations for the authors):It would be useful to specify whether this toolkit covers the following features, and if so, how?– Designing multi-area implants: does the first section (generating surgical coordinates) include the possibility to design multi-probe implants that could target different brain areas? this can be a great value to those who'd like prototype headstages for targeting multiple areas.

We have added this feature in the newest version of HERBS—to accommodate users wishing to target multiple brain areas with a custom-developed multi-probe implant. The user can specify the geometry of the probe following the steps in the illustrated tutorial provided on the GitHub sub-page ( https://github.com/Whitlock-Group/HERBS/blob/main/Tutorial/Probe_Related/5-Design_multi_shanks_Probes.md.)

In addition, the user can define the geometry and site layout on individual probes following a separate illustrated tutorial on the GitHub sub-page linked here. This functionality is noted in the manuscript on lines 113-116, and all probe-related instructions and links to tutorials can be found in the section 6.6 of the CookBook.

– Adaptable to other atlases: is it possible to upload brain atlas for similar animals (e.g. naked mole rats or tree shrews)? How about other available rat brain atlases?

We have added a new feature by which users can upload and use any volume atlas—as long the volume atlas data file (i.e. scanned or imaged brain volume data, such as MRI scans) and the attendant segmentation and label files are provided. Some atlases also include mask files (e.g. the Waxholm Space Atlas), which HERBS can also use.

This functionality is noted in the revised Abstract, and illustrated instructions on how upload and process other atlases can be found on the HERBS GitHub sub-paged linked here. The update and relevant tutorials are noted in subsection 6.1.3 in the Herbs CookBook.

– Integrating other available packages: it would be great if one could easily integrate e.g. CellFinder with HERBS.

We have now made it possible to load cell point data generated from CellFinder, and to visualize those data in 3D brain volumes according to whichever atlas the user has installed. Instructions on how to do this can be found on the HERBS GitHub sub-paged linked here. The link to this tutorial is available in sub-section 6.8.3 of the CookBook, and the added CellFinder functionality is noted in lines 161-165 of the “Reconstruction of viral expression” sub-section of the Results.